# Nuclear High Mobility Group A2 (HMGA2) Interactome Revealed by Biotin Proximity Labeling

**DOI:** 10.3390/ijms24044246

**Published:** 2023-02-20

**Authors:** Antoine Gaudreau-Lapierre, Thomas Klonisch, Hannah Nicolas, Thatchawan Thanasupawat, Laura Trinkle-Mulcahy, Sabine Hombach-Klonisch

**Affiliations:** 1Department of Cellular and Molecular Medicine, Faculty of Medicine, University of Ottawa, Ottawa, ON K1H 8M5, Canada; 2Ottawa Institute of Systems Biology, University of Ottawa, Ottawa, ON K1N 6N5, Canada; 3Department of Human Anatomy and Cell Science, Rady Faculty of Health Sciences, College of Medicine, University of Manitoba, CancerCare Manitoba, Winnipeg, MB R3T 2N2, Canada; 4Department of Pathology, Rady Faculty of Health Sciences, College of Medicine, University of Manitoba, CancerCare Manitoba, Winnipeg, MB R3T 2N2, Canada; 5Department of Medical Microbiology & Infectious Diseases, Rady Faculty of Health Sciences, College of Medicine, University of Manitoba, CancerCare Manitoba, Winnipeg, MB R3T 2N2, Canada; 6Research Institute in Oncology and Hematology (RIOH), CancerCare Manitoba, Winnipeg, MB R3T 2N2, Canada

**Keywords:** HMGA2, nuclear, biotin, proximity labeling, BioID, miniTurbo

## Abstract

The non-histone chromatin binding protein High Mobility Group AT-hook protein 2 (HMGA2) has important functions in chromatin remodeling, and genome maintenance and protection. Expression of HMGA2 is highest in embryonic stem cells, declines during cell differentiation and cell aging, but it is re-expressed in some cancers, where high HMGA2 expression frequently coincides with a poor prognosis. The nuclear functions of HMGA2 cannot be explained by binding to chromatin alone but involve complex interactions with other proteins that are incompletely understood. The present study used biotin proximity labeling, followed by proteomic analysis, to identify the nuclear interaction partners of HMGA2. We tested two different biotin ligase HMGA2 constructs (BioID2 and miniTurbo) with similar results, and identified known and new HMGA2 interaction partners, with functionalities mainly in chromatin biology. These HMGA2 biotin ligase fusion constructs offer exciting new possibilities for interactome discovery research, enabling the monitoring of nuclear HMGA2 interactomes during drug treatments.

## 1. Introduction

High Mobility Group AT-hook protein 2 (HMGA2) is a nuclear non-histone DNA-binding protein that is expressed in embryonic tissues [1] and embryonic stem (ES) cells [2], absent in most normal adult cells, and re-expressed in cancer (stem) cells [3,4,5,6]. The presence of this oncofetal stem cell factor directly correlates with the level of malignancy and metastasis in different cancers [3,7,8,9]. Complex and cell type specific positive and negative transcriptional and translational regulation of HMGA2 expression is tightly controlled by proteins [10] and specific micro and circular RNAs [11,12,13,14,15]. Mutations in the 3′ untranslated region of the HMGA2 gene can impair the binding of specific microRNAs, and this contributes to oncogenic transformation [16]. Cells with reduced HMGA2 protein levels show attenuated epithelial-mesenchymal transition (EMT). The LIN28-Let-7-HMGA2 axis is known to increase cellular HMGA2 levels in stem cells [17] and complex regulatory processes, including a protein complex of Hmga2 with transcription factor Otx2, ensure increased Hmga2 and Lin28 protein levels as a critical step for mouse ESCs transitioning into epiblast-like cell states [18,19]. IGFBP1 RNA binding protein and downstream target of HMGA2 [20] may also support HMGA2 functions in ESC, by binding and promoting translation of mRNAs encoding HMGA2 and LIN28 [21]. In breast, prostate, tongue, and gastric cancers, increased Wnt/β-catenin signaling up-regulates HMGA2 protein and this coincides with EMT and increased tumor aggressiveness [22,23,24].

The ability of HMGA2 to interact with both chromatin and proteins is a critical determinant of HMGA2 functional complexity at the DNA-protein interface. HMGA2 utilizes three multi-functional AT-hook domains for the interaction with AT-rich DNA at the minor groove to cause DNA conformational changes and facilitate transcriptional regulation [25,26]. The AT1-3 hooks possess apurinic/deoxyribo-5′-phosphate (AP/dRP) lyase activity, to promote base-excision repair upon chemotherapeutic stress [27], and a nuclear localization motif in AT2 ensures nuclear localization of HMGA2 [25]. The two linker regions that separate the AT hooks and the second AT-hook are important protein interaction sites, and post-translational modifications of the acidic C-terminus can affect HMGA2 DNA- and protein-binding [25,28,29,30]. Known to participate in the formation of transcription factor complexes and enhanceosomes [11,31], HMGA2 is also recruited to DNA repair complexes at DNA damaged sites [32,33]. The interaction of HMGA2 with retinoblastoma protein (Rb1) activates E2F1, which is a driving force in pituitary adenoma formation in HMGA2 over-expressing mice [34,35,36]. The contributions of HMGA2 to maintaining genomic stability and enhancing resistance against genotoxic stress include protein interactions with both key DNA damage signaling factors, Ataxia teleangiectasia mutated (ATM) and Ataxia teleangiectasia, and Rad3 related (ATR) [37,38] and TFR2, a key member of the shelterin complex at telomeres [39]. At replication forks (RFs), HMGA2 interacts with the replication proteins RPA and PCNA, which protects stalled RFs from collapsing and facilitates early RF re-start [40]. We recently identified PARP1 as a new interaction partner of HMGA2 and showed that high cellular HMGA2 correlated with increased resistance to the PARP inhibitor Olaparib, which coincided with reduced PARP1 trapping to chromatin under Olaparib [41]. Collectively, the impacts of HMGA2 interactions with specific protein partners on cell survival, genome stability, proliferation, and differentiation programs highlight the clinical relevance of HMGA2 protein interactions in stem cell and cancer biology.

Classical antibody-based affinity purification/mass spectrometry (AP/MS) strategies are limited by the quality of the antibodies used for immunoprecipitation (IP). We have used HMGA2 fused to a promiscuous biotin ligase, which served as a bait in quantitative biotin proximity labeling (BPL), to identify stable or transient protein interaction partners of this architectural chromatin binding factor. When the cells are provided with excess biotin, the ligase generates a reactive biotin derivative that diffuses from its active site and can react with free primary amines of exposed lysine residues, resulting in covalent attachment of biotin to proteins in close proximity (<10 nm) to HMGA2. The biotinylated proteins can then be extracted from cells and captured on a streptavidin affinity matrix for identification by liquid chromatography-tandem mass spectrometry (LC-MS/MS). BioID provides the ability to capture weak/transient interactions that can be lost in standard AP approaches, while the strength of the association of biotin with streptavidin permits efficient high-stringency protein extraction and capture methods that help minimize background contaminants [42]. A wide range of biotin ligases have been isolated from bacteria and engineered for use in BioID experiments, with the ongoing aim of improving labeling efficiency (to increase time resolution) while reducing background labeling and tag size. They range from the original 37 kDa E. coli-derived BirA* and its more active TurboID and miniTurbo variants, to the recently described hyperactive BioID2-derivative ultraID (~20kDa) [43,44,45]. The original BioID2 was derived from A. aeolicus, both for its smaller size relative to BirA* (~28 kDa) and the fact that, unlike BirA*, it does not contain a DNA binding domain that could complicate the analysis of chromatin-associated proteins [46]. MiniTurbo was derived from TurboID for the same reason (N-terminal DNA binding domain removed).

In the present study, we have used both HMGA2-BioID2 and HMGA2-miniTurbo fusion proteins to map the HMGA2 interactome in unchallenged HEK293 cells. The selected targets were validated in MDA231 human breast cancer cells and HEK293 cells. These BPL experiments identified new, and confirmed previously described, HMGA2 partners.

## 2. Results

### 2.1. HMGA2-BioID2 Fusion Protein Biotinylates Nuclear Proteins

The Flag-tagged HMGA2-BioID2 fusion protein, and Myc-tagged BioID2 biotin ligase control, run at the expected sizes on a Western blot when detected in HEK293 lysates using anti-Flag and anti-myc antibodies, respectively (Figure 1a). Providing excess biotin to the cells for 4 h induced distinct protein biotinylation patterns that could be detected by probing cell lysates with streptavidin-HRP on a Western blot (Figure 1b), or by staining fixed cells with fluorophore-tagged streptavidin (Figure 1c). Consistent with HMGA2 being predominantly associated with chromatin, the biotinylation pattern of BioID2-tagged HMGA2 was almost exclusively nuclear. In contrast, the diffuse BioID2 ligase alone non-specifically biotinylated proteins throughout the cell (Figure 1c). The biotinylated protein bands observed in the parental HEK cell lysates (Figure 1b) primarily represent mitochondrial carboxylases (e.g., PC, MCCC1/2) that can bind biotin directly, and were also detected as a weak mitochondrial staining pattern in fixed HEK cells (Figure 1c) [42]. These proteins are known background contaminants found in all BioID experiments and are useful for normalization.

### 2.2. HMGA2 Proximity Labeling Identifies A Unique Nuclear HMGA2 Interactome

For the first HMGA2 interactome mapping experiment, HEK293 cells stably expressing HMGA2-BioID2 were labeled with Heavy SILAC media, and we included two controls (HEK293 parental cells labeled with Light media, and BirA*-expressing cells labeled with Heavy media) (Figure 2a). The parental cells served as a control for sticky abundant proteins that may bind non-specifically to the affinity matrix, and also showed biotinylation of the mitochondrial carboxylases that directly bind biotin. BirA* was used in lieu of BioID2 as the ligase alone control, to mark proteins that are non-specifically biotinylated in this experiment, as we had not yet isolated a stable line that expressed BioID2 alone at a similar level to that of the HMGA2-BioID2 fusion protein. For the second experiment, with BioID2 control cells in place, we flipped the SILAC labeling so that the control cells expressing BioID2 alone were labeled with Heavy media, and the HMGA2-BioID2 cells with Light media (Figure 2b). The graph in Figure 2c plots log2 SILAC ratios calculated for 62 proteins that were detected and quantified in both experiments. The log2 H:M ratio (HMGA2 vs. BirA*) was plotted on the x-axis for experiment 1 and the log2 L:H ratio (HMGA2 vs. BioID2) on the y-axis for experiment 2. The gray lines in the upper right quadrant delineate proteins that were enriched > 2-fold above the ratio of the mitochondrial contaminants (indicated on the graph). There were numerous proteins that were only biotinylated in the presence of HMGA2-BD2 in one or both experiments (which precludes quantification of an H:L ratio). Those that were detected in both BioID2 AP/MS experiments were also considered high confidence interactors.

A caveat of this approach, is that the BioID2 biotin ligase requires several hours for efficient BPL, which limits the ability to study dynamic HMGA2 interactions in the presence and absence of short-term treatments. To address this, we generated HEK293 cell lines stably expressing miniTurbo biotin ligase constructs, that allow BPL studies with significantly shorter labeling times (10–60 min). As expected, the biotinylation pattern for HMGA2-miniTurbo is significantly different from that of miniTurbo alone, and localizes predominantly to the nucleus (Figure 3a–c). Importantly, a SILAC BioID AP/MS experiment with a 1 hr labeling time identified our highest confidence HMGA2 interactors, as shown in the graph in Figure 3d (as compared to the BioID2 graph in Figure 2c).

Lastly, we generated an annotated list of high confidence interactors that were enriched in both the BioID2 and the miniTurbo interactome experiments (Table 1). This reveals an exclusive HMGA2 nuclear interactome and highlights the strength of a quantitative BioID approach for interactome mapping.

### 2.3. Nuclear HMGA2-BioID2 Interactome Confirms Known and Discovers New Interaction Partners

Our proximity labeling experiments confirmed several previously described HMGA2 interaction partners. For example, our previous IP studies had identified the early DNA damage sensor PARP1 as an interaction partner of HMGA2 in human breast cancer cells [41]. PARP1 was among the top candidates in our BPL list of HMGA2 interaction partners (Table 1; Figure 2c; Figure 3d), which provided further validation for the new role of HMGA2 in enhancing PARP1-mediated protein ADP-ribosylation of proteins. The non-homologous end-joining (NHEJ) DNA repair factors XRCC6/Ku70 and XRCC5/Ku80 had also previously been demonstrated to interact with HMGA2 [29,48]. We focused on XRCC6/Ku70 and demonstrated co-precipitation of Ku70 with HMGA2 from MDA231 human breast cancer cell nuclear extracts (Figure 4a). Three additional proteins, predicted by our BPL analyses to be interaction partners of HMGA2, were validated by streptavidin pulldown and Western blot detection using specific antibodies to NUMA, TOP2A, and histone H4 (Figure 3b and Figure 4b). We demonstrated a significant increase in the amount of biotinylated NUMA, TOPO2A, and H4 proteins captured in pulldowns from HEK293-HMGA2-BioID2 protein lysates compared to HEK293-BioID2 protein lysates (Figure 4b), which confirmed the quantitative MS data (Figure 2c). To demonstrate the specificity of the identified HMGA2 interactors, we show a representative background contaminant protein (TP53) that is biotinylated non-specifically by both HMGA2-BioID2 and BioID2 alone (Figure 4b). Our repeated attempts to purify a stable HMGA2-TOP2A complex by IP were unsuccessful, which suggests that their interaction is transient. An advantage of BPL over classical antibody-based co-IP studies is the fact that, in addition to detecting stable protein interactions, BPL can also identify biologically relevant transient (or lower affinity) protein-protein interactions. Future studies will explore the functional relevance of the novel HMGA2 interactions that we have mapped.

### 2.4. Bioinformatics Signature of the HMGA2 Proximity Interactome

Our gene ontology methodology utilized Cytoscape (V3.8) with the ClueGO plug-in (v2.5.7) to assess the involvement of 62 high confidence HMGA2 interaction partners identified by BPL (42 mapped by both BioID2 and miniTurbo and an additional 20 only detected in the BioID2 experiments) in biological processes, cellular components, and molecular functions [49,50]. Protein network associations in HEK293-HMGA2-BioID2 cells only considered protein members which networked with a minimum of three other proteins and 4% of the proteins assigned to a biological pathway. Expectedly, chromatin- and nucleosomal-associated processes constituted a prominent element of the HMGA2 protein interactome, frequently linked to GO networks and Reactome pathway analysis, and included known and new interaction partners (Figure 5a–c and Figure 6a,b). The Reactome pathway analysis provided further insight into the predicted HMGA2-nucelosomal interactions. All four core histones (H2a/2b/3/4) were found biotinylated, as validated for histone 4 (Figure 4b), suggesting interactions of HMGA2 with nucleosomes. This was reflected in the Reactome pathway, which prominently featured nucleosomal and chromatin-associated processes, contributing approx. 75% of all pathways. Post-translational protein SUMOylation (10%) and double stranded DNA repair (7%) related Reactome pathways were diversified in HMGA2 interaction partners (Appendix A). Notably, HMGA2 is a target of SUMOylation itself [51,52].

## 3. Discussion

In the present study, we have for the first time used BPL in combination with SILAC protein labeling and LC-MS/MS detection to identify protein interaction partners of HMGA2. We employed two different biotin ligases, BioID2 and the more active miniTurbo, linked to the C-terminus of HMGA2, and mapped the overlapping HMGA2 interactomes with both. We chose to use the BioID2 and miniTurbo ligases for their smaller size and because they do not have a DNA binding domain, that allows them to readily enter the nucleus by free diffusion [53]. An important caveat of the comparison of our hit list with previously published BioID control datasets [47], is that the Lambert study utilized the larger BirA* ligase and compared its non-specific biotinylation pattern when it was concentrated in the nucleus by fusion to a nuclear localization signal vs. concentrated in the cytoplasm by fusion to GFP (~70 kDa fusion protein). By contrast, our smaller BioID2 and miniTurbo ligases allowed us to compare the biotinylation patterns of these ligases when a significant amount was freely diffusing in the nucleus vs. tethered to HMGA2. Biotinylated proteins were largely confined to the nucleus in HEK293-HMGA2-BioID2/miniTurbo cells. 

Traditional co-IP methods tend to favor the detection of high affinity protein interactions and exclude transient and lower affinity interactions [42]. The covalent attachment of a biotin label to lysine residues on nuclear proteins that are in close proximity (<10 nm) to HMGA2 takes place in live cells, after which, stringent protein extraction and pulldown workflows can be used to efficiently capture the biotinylated proteins. This increases assay sensitivity, because the proteins do not need to remain associated after the labeling takes place. The ability of BPL to capture biologically relevant transient interactors with HMGA2 addresses a critical void in our current view of the HMGA2 interactome. Among the proteins not described as HMGA2 interaction partners in previous co-IP studies [29,48] but repeatedly detected in our BPL assays, independent of the biotin ligase employed (BioID2 or miniTurbo), was PARP1, which we had previously identified as a new HMGA2 interaction partner in human breast cancer cells [41]. HMGA2 enhances the activity of PARP1 and this leads to increased PARylation of PARP1 and other proteins, including HMGA2, which can alter protein interactions. BPL consistently captured HMGA2-PARP1 interactions in non-DNA damaged HEK293 cells, suggesting a role for this heterodimer outside of DNA repair. 

HMGA2 protein interactions are anticipated to be cell type- and context-specific, and involve HMGA2 structural changes upon binding to chromatin [54,55]. HMGA2-chromatin interactions occur with distinct affinities to specific DNA conformations, and this may also impact transient HMGA2-protein interactions. To avoid artificial transient protein interactions triggered by high cellular expression levels of HMGA2-biotin ligase fusion protein in our BPL studies, we selected HEK293 stable transfectants that expressed HMGA2-BioID2 and HMGA2-miniTurbo at low to medium levels, comparable to the HMGA2 protein levels detected in cancer cells. Previous proteomic interactome studies have mainly utilized IP of HMGA1, followed by SDS-PAGE protein separation and MS analysis, in HEK293 and human cancer cells. Potential molecular interactors of HMGA1 included mRNA processing factors, RNA helicases, DNA repair factors, and protein chaperons [29]. Employing a phage display screening strategy with an ORF-enriched cDNA library identified several HMGA1 interacting clones that encoded factors involved in transcriptional regulation and chromatin remodeling dynamics [56]. Nuclear putative interaction partners of HMGA2 that were identified in this BPL study in HEK293 cells, cultured under normal conditions, also included chromatin-associated proteins and DNA repair molecules. The latter included the early DNA damage sensor PARP1. Upon treatment with the PARP inhibitor Olaparib, HMGA2 reduced PARP1 trapping at genomic DNA, and this increased the resistance to PARP1 inhibitors [41]. In agreement with data from the Biogrid database, both XRCC5/Ku80 and XRCC6/Ku70 were detected and quantified as possible HMGA2 interaction partners in our BPL assays (https://thebiogrid.org/interaction/869806, (accessed on 29 January 2023); Table 1). We confirmed the interaction of HMGA2 with XRCC6/ Ku70 by co-IP in human MDA231 breast cancer cells (Figure 4). As a single-stranded DNA-dependent ATP-dependent helicase, the XRCC5/6 dimer forms the regulatory subunit of the DNA-dependent protein kinase complex DNA-PK, that is critically important for non-homologous end joining (NHEJ). XRCC6 recognizes broken DNA ends, while the XRCC5/6 DNA helicase dimer blocks further DNA resection, joins loose broken DNA ends, and significantly increases DNA binding and activity of the catalytic subunit PRKDC to aid ligation [57]. Both, the XRCC5/6 complex and HMGA2 display 5’-deoxyribose-5-phosphate lyase (5’-dRP lyase) activity [27,58] and, like HMGA2 [40], XRCC5/6 preferentially binds to forked DNA structures with increased affinity [59,60,61]. XRCC6 has also been identified as an interaction partner of PARP1 in co-IP and BPL studies, and ADP-ribosylation by PARP1 decreases the affinity of Ku for double strand breaks [62,63,64,65]. This XRCC6-PARP1 complex was shown to recruit additional factors involved in DNA maintenance and repair functions, including Werner’s syndrome protein and C/EBP alpha, which may increase the cellular readiness of DNA damage response [66,67]. As a member of the PARP1-XRCC5/6 interactome, HMGA2 may affect complex stability and functionality through its ability to increase PARP1 mediated ADP-ribosylation of proteins, which also may impact on multifactorial PARP1-XRCC5/6 complexes [41]. Finally, stable protein interactions of XRCC6 with PARP1, thyroid receptor-interacting protein 13 (TRIP13), and HMGA1 (a close structural relative of HMGA2) formed in the absence of DNA, but DNA could enhance the protein-protein interaction [68,69,70].

Our BPL analyses identified biotinylated TOPO-2 as a putative HMGA2 interaction partner, but not TOPO-1, as shown by antibody-based co-IP [71,72]. Topoisomerases I and II (TOPO-1/2) have both been implicated as HMGA2 interaction partners, and HMGA2 was shown to antagonize the TOPO-1 antagonist irinotecan/ SN-38 and synergize with TOPO-2 to antagonize topoisomerase-2 inhibitors, such as etoposide or merbarone [72,73]. The HMGA2-TOPO-2A interaction was likely transient (or lower affinity) since our attempts failed to detect HMGA2-TOPO-2A complexes by co-IP. Although outside of the scope of this BPL study, future work may use BPL to explore the possibility of a more stable HMGA2-TOPO-2A complex in the presence of TOPO-2 poisons.

The basic structure of the nucleosome consists of an octameric histone core with two copies each of histones H2A, H2B, H3, and H4 isoforms [74]. These four replication-sensitive core histones were the top four quantified biotinylated proteins in HEK293-HMGA2-BioID2 and HMGA2-miniTurbo cells, with over 60-fold enrichment for some histones versus background. By contrast, the SILAC ratios were much lower for linker histone H1, which is considered to be a known competitor of HMGA2 at chromatin [75]. Surprisingly little is known regarding the direct binding of HMGA proteins to DNA packaged in chromatin fibers. Several HMGA2 molecules can stably interact with a single nucleosome core [26], and a recent BioID2 BPL core histone interactome mapping provided the first evidence of an HMGA2 interaction, with H3A used as bait [47,76]. The emerging relationship between histones and HMGA2 is complex and suggests that HMGA2 interactions with histones and/or histone modifying proteins may affect accessibility to chromatin. Downregulated or deleted HMGA2 causes developmental delays in embryonic stem cells and coincides with increased chromatin condensation, with a repressive trimethylation of K27 of histone H3 (H3K27me3) signature and increased polycomb repressive complex 2 (PRC2) catalytic activity [77]. Human pancreatic adenocarcinoma cells grown in a collagen matrix showed enhanced acetylation of H3 histones, H3K9ac and H3K27ac, which was catalyzed by specific promoter interactions of HMGA2 causing transcriptional activation of the histone acetyltransferase (HAT) family members GCN5, p300, and PCAF [78]. In hepatocellular carcinoma, an oncogenic EGF-EGFR-PI3K/Akt pathway was shown to increase p300 HAT-mediated H3K9 acetylation of the HMGA2 promoter, resulting in higher HMGA2 gene expression and a poor prognosis [79]. In glioma, a HMGA2/GCN5 protein complex promoted histone acetylation in nucleosomes at AT-rich DNA sites, to cause gene activation of matrix metalloproteinase 2 (MMP2), a mediator of invasiveness in glioma, and conferring a poor prognosis [80]. Overexpression of HMGA2 coincided with elevated levels of H2A, H2B, H3, and H4 core histones in a subpopulation of dedifferentiated liposarcoma, again conferring a poor prognosis [81]. An even more intricate HMGA2-histone relationship is suggested by recent evidence that demonstrates the requirement for HMGA2 to induce DNA nicks to facilitate the incorporation of phosphorylated H2A.XS139 isoform (γ-H2AX) into nucleosomes to earmark DNA sites for repair-mediated DNA demethylation and transcription activation [33]. In addition, histones may also facilitate HMGA2 interactions with specific chromosomal regions. An earlier report had identified HMGA2 at centromeres and telomeres [82]. We had previously identified the shelterin protein TRF2 as a telomeric binding partner of HMGA2 [39], and epigenetic centromeric marker histone 3 variant CENP-A may direct HMGA2 to centromeres [83]. A possible reason why we did not detect TRF2, despite demonstrating the direct interaction of this shelterin protein with HMGA2 previously [39], may be the fact that biotinylation relies not only on proximity to the cloud of activated biotin generated by the ligase but also on the availability of exposed lysine residues to which biotin can covalently attach. This highlights the importance of complementary approaches for interactome mapping, and follow-up analyses of our BioID high-confidence hits will include a range of different techniques.

The nucleolus is an organelle considered a driver of cancer progression [84]. One study in maize identified Hmga2-GFP localized preferentially to nucleoli as the site of ribosomal biogenesis [85]. We have demonstrated an interaction of HMGA2 with the nuclear mitotic apparatus protein, NUMA. This multifunctional chromatin organization factor [86,87] is essential for organizing the nuclear matrix at spindle poles, and serves as a co-activator of p53 dependent pathways [88,89], DNA repair [90,91] and apoptosis [92]. Because we detected NUMA biotinylation with both HMGA2-BioID2 and, the much shorter, HMGA2-miniTurbo labeling, we may have labeled nucleolar NUMA. NUMA regulates rRNA levels through complex interactions with ribosomal proteins and RNA to preserve ribonuclear protein network integrity [86,87,93]. Intriguingly, NUMA is emerging as an activator of p53 independent nucleolar stress pathways, and the observed biotinylated NUMA in our BPL analysis of HEK293-HMGA2-BioID2/miniTurbo cells may suggest novel nucleolar roles of HMGA2 [94].

In conclusion, we have demonstrated the feasibility of BPL to unveil nuclear HMGA2 interactomes, which confirmed existing, and revealed new, HMGA2 interaction candidates. Our HMGA2 BPL study firmly ascertained the role of HMGA2 as a chromatin binding protein but also revealed new relationships with nucleosomes and the nucleolus. Their functional relevance for normal and neoplastic cell functions, and/or therapeutic targeting quality for the treatment of HMGA2+ cancers, continues to be the topic of ongoing studies.

## 4. Materials and Methods

### 4.1. Creation of Stable HEK293 Cell Lines Expressing BioID2 and Miniturbo Fusion Proteins

Using random integration, we generated stable HEK293 cell clones overexpressing low levels of either, the biotin ligases alone (myc-BioID2 or Flag-miniTurbo), or the ligases fused to the C-terminus of HMGA2 (HMGA2-Flag-BioID2 or HMGA2-V5-miniTurbo). All expression plasmids used to generate the stable lines were sequenced to confirm their identity, and protein expression was confirmed by Western blot analysis (Figure 1 and Figure 3).

### 4.2. Metabolic Labeling and Liquid Chromatography Mass Spectrometry (LC-MS/MS) Analysis

To eliminate the variability between individual control and experimental mass spectrometry (MS) runs, and to improve our ability to distinguish specific but low abundance interactors from the larger number of non-specifically biotinylated and abundant sticky proteins, we used stable isotope labeling by amino acids in culture (SILAC)-based metabolic labeling. This approach allowed us to combine the proteins captured from separate control and HMGA2 experiments prior to elution from the affinity matrix and subsequent MS analysis (Figure 2a,b). The amount of protein captured in the experimental vs control condition was thus determined in a single MS run. Briefly, cells were differentially labeled with media containing either, environmental forms of the essential amino acids Arginine and Lysine (Arg0Lys0, Light media), or isotopic forms (Arg6Lys4, Medium; Arg10Lys8, Heavy) (Cambridge isotope Laboratories, Tewksbury, MA, USA; Athena ES, Baltimore, MD, USA). Biotin (Sigma-Aldrich, St. Louis, MO, USA) was added in excess to the media (50 μM) and the cells were harvested 18 h later by scraping them into an ice-cold high-salt RIPA buffer (50 mM Tris pH 7.5, 500 mM NaCl, 1% NP-40, 0.5% deoxycholate, protease inhibitors), sonicating, and clearing by centrifuging at 21,000× *g* for 10 min at 4 °C. Total protein concentrations were measured using the Pierce BCA Protein Assay Kit (Thermo-Fisher, Waltham, MA, USA). The salt concentration in the whole cell extracts was then reduced to 250 mM by adding an equal volume of “no salt” (0 mM NaCl) RIPA buffer, and equivalent amounts of total protein extract for each condition were incubated with Streptavidin-agarose beads (Thermo-Fisher, Waltham, MA, USA) at 4 °C for 4 h. Following an initial wash with RIPA buffer, beads from the control and experimental pulldowns were combined for additional washes and bound proteins eluted with 2% SDS/30 mM biotin. The eluted proteins were reduced and alkylated by treatment with DTT and iodoacetamide, respectively. Sample buffer was then added and the proteins resolved by electrophoresis on a NuPAGE 10% BisTris gel (Thermo-Fisher, Waltham, MA, USA). The gel was stained using SimplyBlue Safestain (Thermo-Fisher, Waltham, MA, USA) and the entire lane was cut into five slices. Each slice was cut into 2 × 2 mm fragments, destained, and digested overnight at 30 °C with Trypsin Gold (Thermo-Fisher, Waltham, MA, USA).

An aliquot of each tryptic digest was analyzed by LC-MS/MS on an Orbitrap Fusion Lumos system (Thermo-Fisher, Waltham, MA, USA) coupled to a Dionex UltiMate 3000 RSLC nano HPL. The raw files were searched against the Human UniProt Database using MaxQuant software v1.5.5.1 (http:/www.maxquant.org accessed on 29 January 2023) [95] and the following criteria: peptide tolerance = 10 ppm, trypsin as the enzyme (two missed cleavages allowed), and carboxyamidomethylation of cysteine as a fixed modification. Variable modifications are, oxidation of methionine and N-terminal acetylation. Heavy SILAC labels were Arg10 (R10) and Lys8 (K8). Quantitation of SILAC ratios was based on razor and unique peptides, and the minimum ratio count was 2. The peptide and protein FDR was 0.01. These ratios reflected the relative amount of protein that was captured in experimental (biotin ligase-tagged HMGA2) vs. control (biotin ligase alone) experiments. 

### 4.3. Western Blot Validation by Co-Immunoprecipitation (IP) or BioID

For co-immunoprecipitation (co-IP) of HMGA2 and XRCC6/Ku70, we used nuclear extracts from MDA-MB-231 breast cancer cells. The cytoplasmic fraction was first separated using the NE-PER kit (Thermo Fisher, Waltham, MA, USA), according to the manufacturer’s instructions. The nuclear pellet was lysed using the lysis buffer containing 50 mM Tris-HCl (pH 7.5), 150 mM NaCl, 25 mM NaF, 0.1 mM Na3VO4, 0.2% Triton X-100, 0.3% NP-40 and protease inhibitors. The mixture was incubated on ice for 40 min with intermittent vortexing every 10 min, followed by centrifugation at 16,000× *g* for 10 min at 4 °C. Supernatant containing the nuclear fraction was used for IP. Nuclear extracts (100 µg) were incubated with 0.6 µg HMGA2 antibody (Cell Signaling Technology, Whitby, ON, Canada) overnight at 4 °C, and then incubated with 100 µL of protein A/G magnetic beads (Thermo Fisher, Waltham, MA, USA) for 4 h with shaking, at 4 °C. The reaction complex was washed three times with lysis buffer and bound proteins eluted in 3x Laemmli buffer. The samples were boiled at 95 °C for 5 min prior to SDS-PAGE (Bio-Rad, Mississauga, ON, Canada) and proteins were transferred to a nitrocellulose membrane. For immunodetection, nonspecific protein binding sites were blocked with 5% non-fat milk in TBS/T (0.1% Tween 20 in Tris-buffered saline). Primary antibodies (1:1000 dilutions of anti-HMGA2 or anti-Ku70 [both Cell Signaling Technologies, Whitby, ON, Canada]) were incubated at 4 °C overnight. Membranes were washed three times in TBS/T for 5 min each at RT before incubating with HRP-conjugated goat anti-rabbit IgG secondary antibody (Cell Signaling Technologies, Whitby, ON, Canada) for 1 h at RT. Immunoreactive bands were visualized with ECL Clarity (Bio-Rad, Mississauga, ON, Canada) using Bio-Rad Chemi-Doc MP Imagers. For the BioID/Western blot validation experiments, the samples were prepared as described for the MS workflow up to the SDS-PAGE gel separation step, after which the proteins were transferred from the gel to nitrocellulose for Western blot analysis. This was carried out as described above, with the exception that the membranes were incubated with primary antibodies (all 1:1000 dilutions) for 1 h at room temperature.

### 4.4. Bioinformatics Analysis

UniProt IDs and Entrez GeneIDs were used for the network and pathway analyses. Cytoscape (version 3.8) with the ClueGO V2.5.9 plug-in was used for Gene Ontology (GO) enrichment analyses and identification of Reactome pathways in the list of high confidence HMGA2 interactors. The ClueGO V2.5.9 plug-in generates functionally grouped GO annotation networks from a large cluster of genes. GO categories included biological process, cellular component, and molecular function terms. Our criteria for creating networks were a minimum of three genes and at least 4% of the genes being assigned to a biological pathway. *p*-values were calculated using the hypergeometric test and adjusted for multiple testing with the Benjamini–Hochberg method. Adjusted *p*-values < 0.05 were considered statistically significant as denoted by ** *p* < 0.001, * *p* < 0.01, without star *p* < 0.05.

## 5. Conclusions

We have demonstrated the feasibility of BPL to unveil nuclear HMGA2 interactomes, which confirmed existing, and revealed new, HMGA2 interaction candidates. Our HMGA2 BPL study firmly ascertained the role of HMGA2 as a chromatin binding protein, but also revealed new relationships with nucleosomes and the nucleolus. Their functional relevance for normal and neoplastic cell functions, or therapeutic targeting quality, is the topic of ongoing studies.

## Figures and Tables

**Figure 1 ijms-24-04246-f001:**
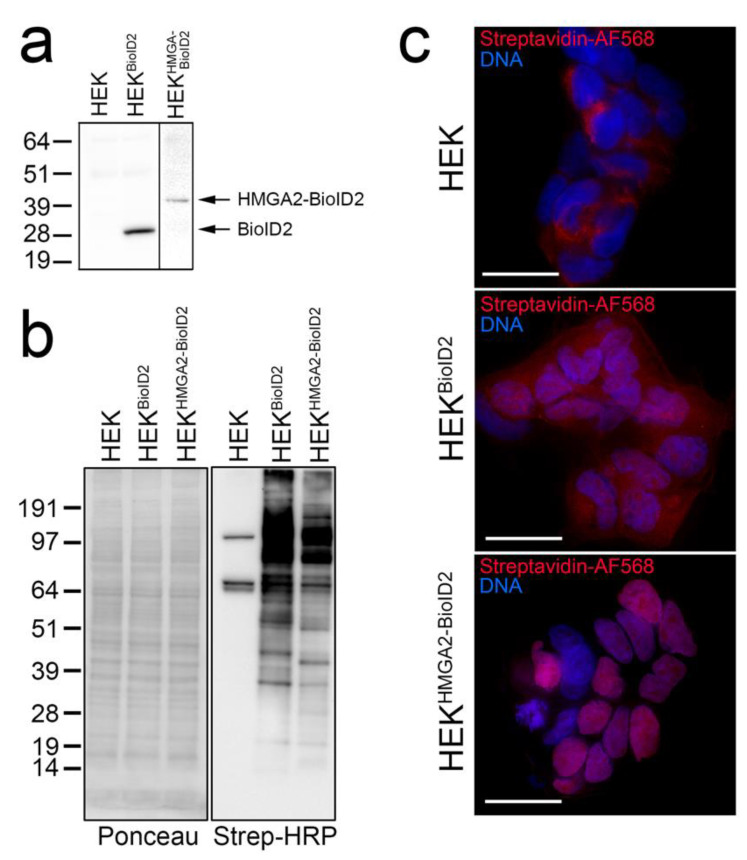
(**a**–**c**) Validation of BioID2 stable cell lines for BioID. (**a**) Western blot analysis of the apparent MW of myc-tagged BioID2 and Flag-BioID2-tagged HMGA2 stably expressed in HEK cells and detected using anti-myc (Millipore Sigma, Burlington, MA, USA) or anti-Flag (Millipore Sigma, Burlington, MA, USA) antibodies, respectively. (**b**) Western blot analysis of whole cell lysates (50 μg total protein loaded) harvested from the HEK parental cell line and HEK-BioID2 and HEK-HMGA2-BioID2 stable cell lines after 4 h labeling with 50 μM biotin. Ponceau S staining of the membrane after transfer is shown on the left, and Streptavidin-HRP (Thermo-Fisher, Waltham, MA, USA) detection of biotinylated proteins on the right. (**c**) Cells grown on coverslips in the same dishes were formaldehyde-fixed, permeabilized, and stained with Streptavidin-AlexaFluor 568 (Thermo-Fisher, Waltham, MA, USA) (red; biotinylated proteins) and Hoechst 33342 (Millipore Sigma, Burlington, MA, USA) (blue; DNA). Scale bars are 20 μm.

**Figure 2 ijms-24-04246-f002:**
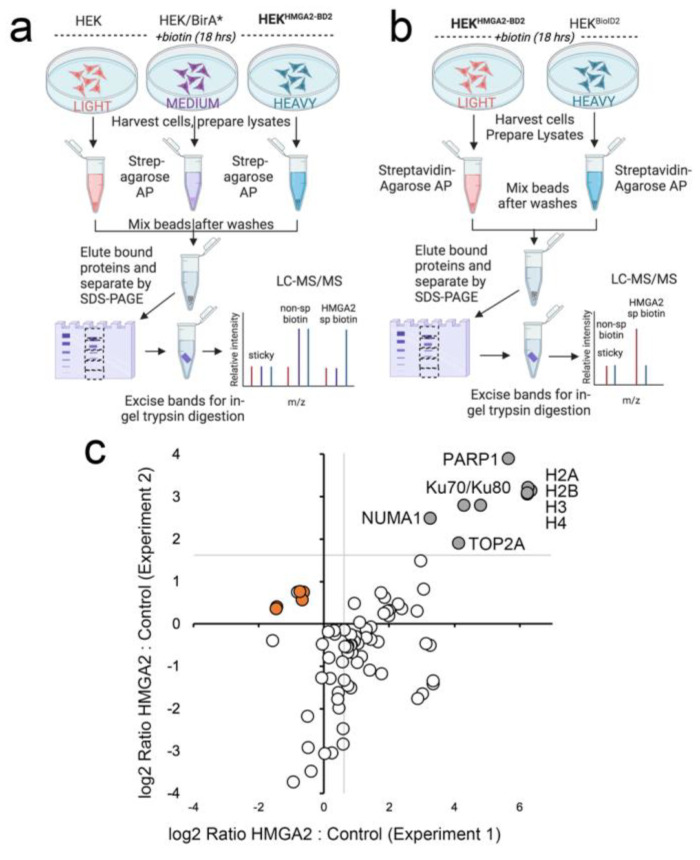
(**a**–**c**) SILAC BioID2 experiments identify high-confidence interactors for HMGA2. (**a**) Design of the first SILAC BioID experiment, in which the HEK-HMGA2-BioID2 cells were labeled with Heavy Arg10Lys8 media. Two controls were included: cells transiently overexpressing ligase alone (in this case BirA*), to highlight proteins that are non-specifically biotinylated (labeled with Medium Arg6Lys4 media); and parental cells, to highlight sticky proteins (labeled with Light Arg0Lys0 media). (**b**) Design of the second SILAC BioID experiment, in which the labeling was flipped so that the HEK-HMGA2-BioID2 cells were grown in Light R0K0 media and the control ligase alone cells (in this case, HEK-BioID2 stable cells) in Heavy R10K8 media. For both experiments, cells were provided with 50 μM biotin for 18 hrs prior to harvesting (**c**) Graph plotting the log2 Ratio L:H (HMGA2-BD2:BD2) for experiment 2 vs. the log2 Ratio H:M (HMGA2-BD2:BirA*) for experiment 1 for 78 proteins that were detected and quantified in both. Proteins enriched > 2-fold above background in both experiments are indicated (gray circles) and considered high-confidence interactors. Mitochondrial carboxylases that bind biotin directly and represent background contaminants are shown as orange circles.

**Figure 3 ijms-24-04246-f003:**
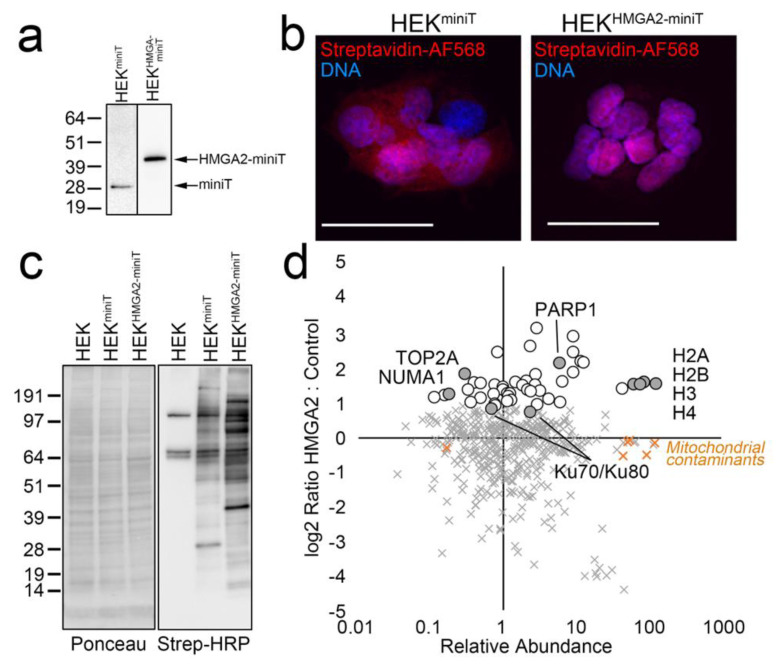
(**a**–**d**) Validation of miniTurbo stable cell lines for BioID with shorter labeling time. (**a**) Western blot analysis of the apparent MW of Flag-tagged miniTurbo and V5-miniTurbo-tagged HMGA2 stably expressed in HEK cells and detected using anti-Flag (Millipore Sigma, Burlington, MA, USA) or anti-V5 (Thermo-Fisher, Waltham, MA, USA) antibodies, respectively. (**b**) Cells grown on coverslips in the same dishes were fixed, permeabilized and stained with Streptavidin-AlexaFluor 568 (red; biotinylated proteins) and Hoechst 33342 (blue; DNA). Scale bars are 20 μm. (**c**) Western blot analysis of the HEK parental cell line and HEK-miniT and HEK-HMGA2-miniT stable cell lines after 1 hr labeling with 50 μM biotin. Ponceau S staining of the membrane after transfer is shown on the left, and Streptavidin-HRP detection of biotinylated proteins on the right. (**d**) Graph plotting the log2 Ratio H:L (HMGA2-miniT: miniT) vs. relative abundance for all proteins identified and quantified in a SILAC miniTurbo BioID experiment. Proteins enriched >2-fold above the background are indicated (white circles), as are the high-confidence interactors identified in both BioID2 experiments (gray circles). Mitochondrial carboxylases that bind biotin directly are also shown (orange X).

**Figure 4 ijms-24-04246-f004:**
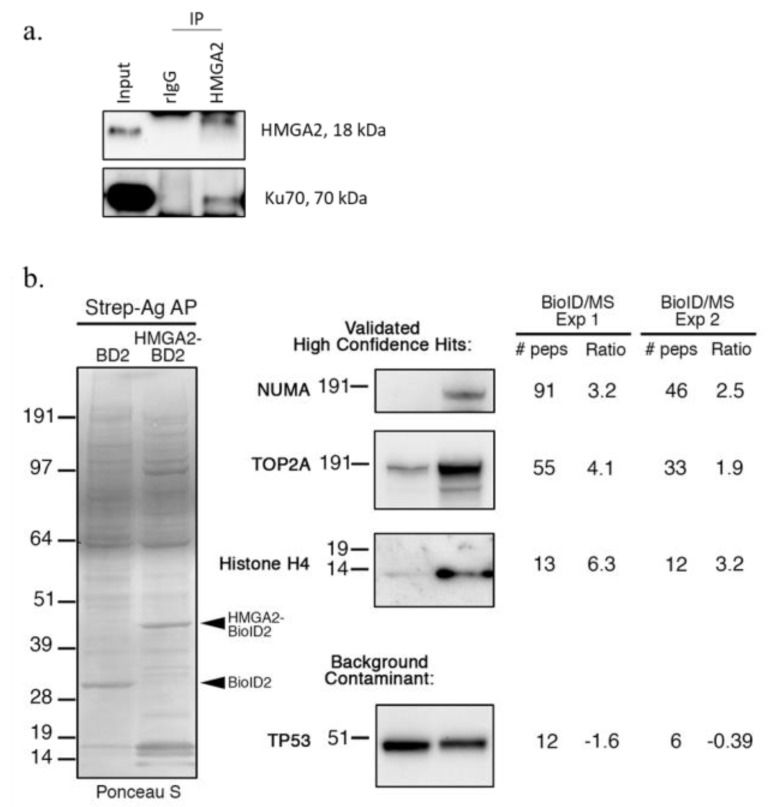
(**a**,**b**) Validation of candidate HMGA2 interaction partners. (**a**) Immunoprecipitation (IP) analysis of HMGA2 interaction with XRCC6 in MDA-MB-231 cells. Nuclear extracts were used for IP of HMGA2 with anti-HMGA2 antibodies. Rabbit IgG (rIgG) was used as a negative IP control. Immunoblotting confirmed co-precipitation of XRCC6 with HMGA2. (**b**) BioID/Western blot of analysis of proteins biotinylated by BioID2 alone (BD2) vs. BioID2-tagged HMGA2 (HMGA2-BD2) in the HEK293 stable cell lines in the presence of excess biotin for 18 h. Biotinylated proteins were captured on Streptavidin-agarose (Strep-Ag) beads (Thermo-Fisher, Waltham, MA, USA). Total proteins were first detected by Ponceau S staining of the membrane, and then NUMA (New England Biolabs, Ipswich, MA, USA), TOP2A (New England Biolabs, Ipswich, MA, USA), Histone H4 (Abcam, Waltham, MA, USA), and TP53 (Santa Cruz Biotechnologies Inc, Dallas, TX, USA) detected using antibodies raised against each protein. For each protein, the number of peptides detected (and log2 SILAC Ratio) in both BioID2 AP/MS experiments is indicated. TP53 was included as an example of a background contaminant (SILAC ratio ≤ 1:1) that is non-specifically biotinylated by both HMGA2-BioID2 and BioID2 alone.

**Figure 5 ijms-24-04246-f005:**
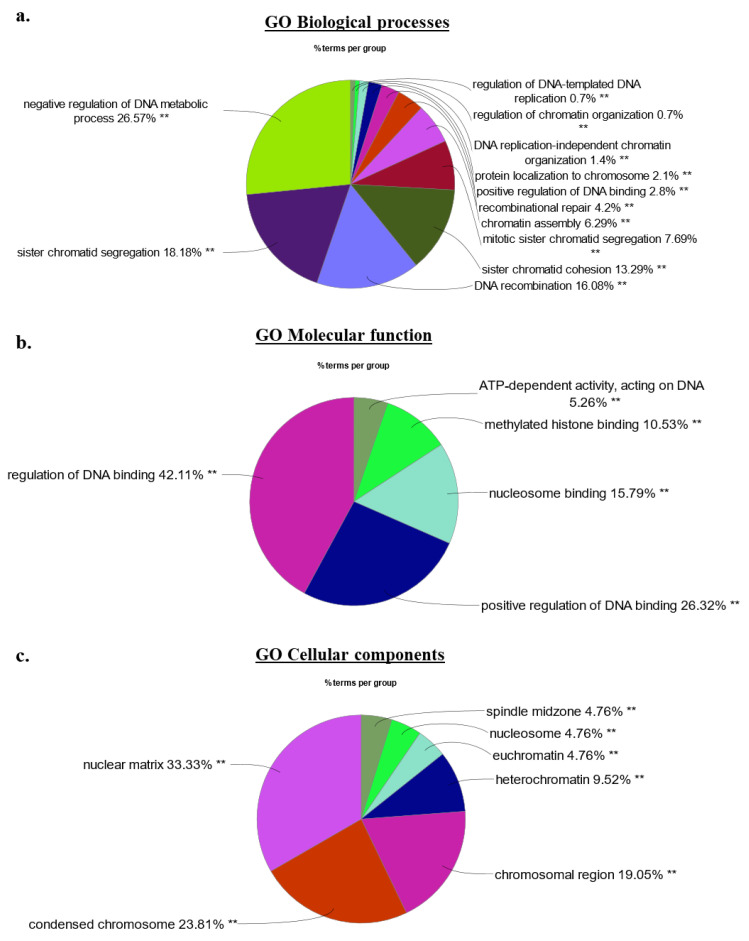
(**a**–**c**) Gene ontology analysis of the 62 top candidate proteins identified in the HMGA2-BioID2 and -miniTurbo biotin proximity assays. Nuclear chromatin- and nucleosome associated processes and functions constitute the main GO terms associated with the top 62 HMGA2 nuclear interaction candidates identified by biotin proximity labeling in unchallenged HEK293 cells. Percentage distribution of major (**a**) GO–Biological processes, (**b**) GO–Molecular functions, and (**c**) GO–Cellular components, as determined by Cytoscape version 3.8 and ClueGo v2.5.9. Adjusted *p*-values considered statistically significant as denoted by ** *p* < 0.001.

**Figure 6 ijms-24-04246-f006:**
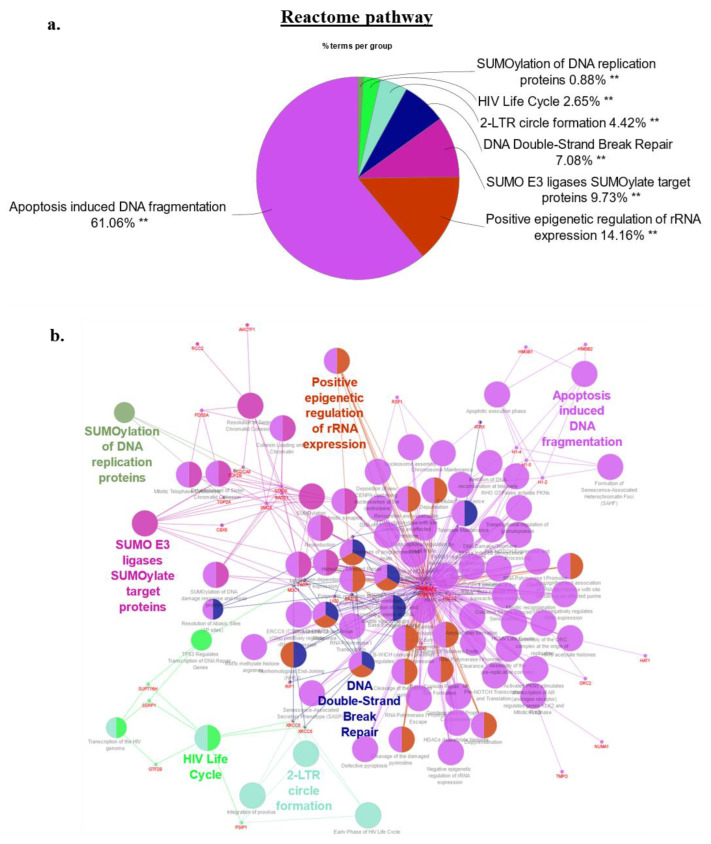
(**a**,**b**)REACTOME pathway analysis of the 62 top candidate proteins identified in the HMGA2-BioID2 and -miniTurbo biotin proximity assay. (**a**) There were 7 REACTOME pathway groups identified, with the two largest groups 5 and 6 mainly encompassing nucleosomal and chromatin-associated processes, and contributing approx. 75% of all pathways due to the over-representation of nucleosomal histone interaction partners of HMGA2 in both pathways. Pathways with more diversified contributions by HMGA2 interaction partners included post-translational protein SUMOylation related REACTOME pathways, accounting for 10%, and double stranded DNA repair (7%). A detailed list of contributing HMGA2 interaction partners to the Reactome pathways is presented in Appendix A. (**b**) Network map with identical color coding of key Reactome pathways. Analysis was performed with Cytoscape version 3.8 and ClueGo v2.5.9. Adjusted *p*-values considered statistically significant as denoted by ** *p* < 0.001.

**Table 1 ijms-24-04246-t001:** Top HMGA2 interacting candidates identified in both SILAC BioID2 and miniTurbo AP/MS experiments. For each experiment, the # of peptides identified is noted, along with the log2 SILAC ratio quantified by MaxQuant or calculated based on intensity values (italicized). “Bait only” indicates that the protein was only biotinylated/captured from cells expressing ligase-tagged HMGA2. Proteins annotated on Biogrid as putative HMGA2 interactors are indicated. Comparison to the Gingras lab’s BioID control dataset [47] is annotated as follows: (-) not detected, (≤) fewer or same number of peptides detected in NLS-BirA* datasets compared to GFP-BirA* datasets, (# peptides detected in NLS-BirA* datasets/# peptides detected in GFP-BirA* datasets). Proteins in bold text were further validated by co-precipitation or BioID/Western blot analysis (see Figure 4).

		BioID2 AP/MS				miniTurbo AP/MS		
		Experiment 1		Experiment 2				
UniProt	Gene	# pep	log2 Ratio	# pep	log2 Ratio	# pep	log2 Ratio	Biogrid
Bait								
FSH2U8	HMGA2	6	Bait only	4	Bait only	3	Bait only	
Histones								
O75367	H2AY	13	8.52	9	2.95	4	1.28	*
P16403	HIST1H1C	11	2.96	12	1.49	12	0.68	
P10412	HIST1H1E	9	Bait only	11	1.46	11	0.42	
Q93077	HIST1H2AC	6	5.5	7	3.16	5	1.55	*
Q99878	HIST1H2AJ	8	6.22	8	3.22	5	1.57	
**P62805**	**HIST1H4F**	**13**	**6.34**	**12**	**3.15**	**10**	**1.58**	
Q71DI3	HIST2H3D	9	6.23	11	3.06	4	1.63	
Q99880	H2B1L/M	11	6.22	9	3.1	4	1.56	
Other Hits								
P25440	BRD2	10	Bait only	5	Bait only	7	Bait only	
Q13185	CBX3	6	Bait only	6	1.07	7	1.38	
E9PEI0	CDCA2	4	Bait only	1	1.73	7	3.26	
Q53HL2	CDCA8	6	Bait only	2	2.53	10	5.55	
Q96JM3	CHAMP1	18	Bait only	14	1.34	13	1.41	
Q5QJE6	DNTTIP2	8	Bait only	3	2.05	7	1.41	
P78347	GTF2I	26	Bait only	16	1.59	20	0.46	
O14929	HAT1	3	Bait only	4	2.48	1	Bait only	
Q5T7C4	HMGB1	6	3.05	4	0.82	5	1.85	
D6R9A6	HMGB2	5	4.13	4	0.82	4	1.66	
Q9H0C8	ILKAP	11	Bait only	9	1.8	5	0.85	
P42167	LAP2B	19	Bait only	22	1.08	9	2.19	
Q14676	MDC1	32	Bait only	39	2.61	5	0.62	
P46013	MKI67	4	Bait only	83	1.62	106	2.51	
**Q14980**	**NUMA1**	**91**	**3.25**	**46**	**2.49**	**12**	**1.29**	
P09874	PARP1	66	5.65	54	3.9	24	2.19	*
Q7Z3K3	POGZ	15	Bait only	6	4.23	4	1.19	
O75475	PSIP1	16	Bait only	7	3.02	3	Bait only	*
Q08945	SSRP1	17	Bait only	8	1.48	2	1.26	
**P11388**	**TOP2A**	**55**	**4.13**	**33**	**1.9**	**15**	**1.86**	
Q9ULW0	TPX2	20	Bait only	20	1.56	23	2.96	
Q14669	TRIPC	25	Bait only	14	2.37	20	1.59	
B7ZM82	WIZ	13	Bait only	9	1.95	11	Bait only	
P13010	XRCC5	32	4.13	20	2.8	10	0.86	*
**P12956**	**XRCC6**	**29**	**4.29**	**20**	**2.8**	**10**	**0.75**	*****
P17028	ZNF24	6	Bait only	3	Bait only	2	Bait only	
Q96KM6	ZNF512B	3	Bait only	5	2.57	2	Bait only	

## Data Availability

The authors will make the MS data available upon reasonable request.

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
