# Peer review of "Nuclear High Mobility Group A2 (HMGA2) Interactome Revealed by Biotin Proximity Labeling"

_ijms, 2023, doi:10.3390/ijms24044246_

Round 1

Reviewer 1 Report

In this manuscript the authors report the findings of a screen for HMGA2 interaction partners utilizing biotin proximity ligation.

Significance: Identifying novel interaction partners of HMGA2 would greatly aid in further elucidating its biological role

This proteomic screen with cell lines carefully selected to not overexpress the bait fused to the enzyme (BioID2 or miniTurbo) should be published after the authors address the (relatively straightforward) points below.

Comments:

Main points to be addressed:

1)      Designing appropriate control baits for biotin proximity ligation is complicated. In that regard, using a biotin ligase that mostly localizes to the cytosol (Fig. 1) as control was not an optimal choice. Maybe a fusion protein with a nuclear localization signal would have been better. This would serve as a control for – especially highly abundant - nuclear proteins (histones, etc.) being labelled by chance for being in the “cloud” of activated biotin. Yet a redesign of the screen would not be reasonable at this point. I therefore suggest using the results of previously published BioID screens with nuclear proteins for comparison. The lab of Dr. Anne-Claude Gingras might have published data on “commonly found background nuclear proteins” in their many screens.

2)      In the abstract, the authors discuss validation in both MDA-MB-231 (TYPO in abstract!) and HEK293, but only data from MDA-MB-231 are shown. Please include those in Fig. 4. Furthermore, I strongly would suggest in the co-IP to include experiments with nuclease or EtBr to test whether the interactions are DNA mediated.

3)      The discussion could include drawbacks of biotin proximity ligation. Especially the mentioning of TRF2 (identified as HMG2A binding partner previously by the lab, Natarjan et al., Oncotarget 2016), which seems not to feature in this screen, would provide the opportunity to discuss the amino acid dependency of biotin ligation (and the high likelihood of biotinylation of proteins such as histones) as well as a comparison to APEX proximity labelling.

Minor points:

In the introduction the regulation of HMG2A is extensively described. It is not clear why this is relevant, as the authors have not identified (or discussed) a regulator of HMG2A in their screen.

Fig. 3 is mislabelled (b versus c)

There are a few spelling mistakes (e.g., “formaldehyde” in Fig.1.)

Author Response

Reviewer 1
We thank reviewer 1 for their insightful comment on the Gingras work which we have cited in the revised manuscript. We also amended Table 1 to reflect the findings by Gingras and enable readers to
Page 2 of 2
directly compare the data from our HMGA2 screen with that of the Gingras team. We again thank the reviewer for suggesting this addition.
We have corrected typos throughout the manuscript and as indicated by this reviewer.
We have added new citations to the manuscript which specifically investigated the role of genomic DNA for the immunoprecipitation of XRCC6 complexes with other proteins, including HMGA1. In all three protein complexes investigated, DNA encouraged complex formation but was not critical for the protein-protein interactions. We included this statement in the revised manuscript. Our own co-IP studies with HMGA2-PARP1 are in agreement with the above findings as well (Hombach-Klonisch et al., Mol Oncol 2019, 153-170).
In the Discussion part, I added a few sentences to highlight caveats when performing and interpreting biotin proximity ligation protein interaction studies. We also addressed the comment on the lack of detection of telomeric TRF2 protein in our screen.
Minor concerns were addressed and have been remedied.

Reviewer 2 Report

In this publication, Gaudreau-Lapierre et al used BPL in combination with SILAC protein labeling and LC-MS/MS detection to identify protein interaction partners of HMGA2. They present the advantages of this new method over the traditional co-IP methods, validate some HMGA2 previously described interactome, and identify new partners for this protein.  

I believe that the research design is appropriate, the results are clearly presented, and the methods are adequately described.

I would recommend increasing the font type in figure 5 and for sure improving the quality of figure 6. The Reactome presented in figure 6b is not very clear and most of the text cannot be read.

Author Response

As requested by reviewer 2, we have shortened the part in the Introduction which relates to the regulation of HMGA2.
We also remedied the typos and corrected Fig. 3 legend to adjust for the mislabelled Fig. 3.